# Impact of etonogestrel implant use on T-cell and cytokine profiles in the female genital tract and blood

Lisa B. Haddad[1]*, Alison Swaims-Kohlmeier[2], C. Christina Mehta[3], Richard E. Haaland[2], Nakita L. Brown[4,5], Anandi N. Sheth[4,5], Hsin Chien[4,5], Kehmia Titanji[6], Sharon L. Achilles[7], Davis Lupo[2], Clyde E. Hart[2], Igho Ofotokun[4,5]

1 Department of Gynecology and Obstetrics, Emory University School of Medicine, Atlanta, Georgia, United States of America, 2 Laboratory Branch, Division of HIV/AIDS Prevention, Centers for Disease Control and Prevention, Atlanta, Georgia, United States of America, 3 Department of Biostatistics and Bioinformatics, Emory University Rollins School of Public Health, Atlanta, Georgia, United States of America, 4 Department of Medicine, Division of Infectious Diseases, Emory University School of Medicine, Atlanta, Georgia, 5 Grady Healthcare System, Atlanta, Georgia, United States of America, 6 Department of Medicine, Division of Endocrinology, Emory University School of Medicine, Atlanta, Georgia, United States of America, 7 Department of Obstetrics, University of Pittsburg, Gynecology and Reproductive Sciences, Pittsburg, Pennsylvania, United States of America

* lbhadda@emory.edu

**Data Availability Statement:** All relevant data are available in OPEN ICPSR (https://www.openicpsr.org/openicpsr/project/118172/version/V1/view?path=/openicpsr/118172/fcr:versions/V1).

## Abstract

### Background

While prior epidemiologic studies have suggested that injectable progestin-based contraceptive depot medroxyprogesterone acetate (DMPA) use may increase a woman's risk of acquiring HIV, recent data have suggested that DMPA users may be at a similar risk for HIV acquisition as users of the copper intrauterine device and levonorgestrel implant. Use of the etonogestrel Implant (Eng-Implant) is increasing but there are currently no studies evaluating its effect on HIV acquisition risk.

### Objective

Evaluate the potential effect of the Eng-Implant use on HIV acquisition risk by analyzing HIV target cells and cytokine profiles in the lower genital tract and blood of adult premenopausal HIV-negative women using the Eng-Implant.

### Methods

We prospectively obtained paired cervicovaginal lavage (CVL) and blood samples at 4 study visits over 16 weeks from women between ages 18–45, with normal menses (22–35 day intervals), HIV uninfected with no recent hormonal contraceptive or copper intrauterine device (IUD) use, no clinical signs of a sexually transmitted infection at enrollment and who were medically eligible to initiate Eng-Implant. Participants attended pre-Eng-Implant study visits (week -2, week 0) with the Eng-Implant inserted at the end of the week 0 study visit and returned for study visits at weeks 12 and 14. Genital tract leukocytes (enriched from CVL) and peripheral blood mononuclear cells (PBMC) from the study visits were evaluated

**Funding:** This study was supported by the U.S. Centers for Disease Control and Prevention, the NIH NICHD K23HD078153-01A1 (L.B.H) and the Emory University Center for AIDS research (P30AI050409).

**Competing interests:** no authors have competing interests.

for markers of activation (CD38, HLA-DR), retention (CD103) and trafficking (CCR7) on HIV target cells (CCR5+CD4+ T cells) using multicolor flow cytometry. Cytokines and chemokines in the CVL supernatant and blood plasma were measured in a Luminex assay. We estimated and compared study endpoints among the samples collected before and after contraception initiation with repeated-measures analyses using linear mixed models.

## Results

Fifteen of 18 women who received an Eng-Implant completed all 4 study visits. The percentage of CD4+ T cells in CVL was not increased after implant placement but the percentage of CD4+ T cells expressing the HIV co-receptor CCR5 did increase after implant placement (p = 0.02). In addition, the percentage of central memory CD4+ T-cells (CCR7+) in CVL increased after implant placement (p = 0.004). The percentage of CVL CD4+, CCR5+ HIV target cells expressing activation markers after implant placement was either reduced (HLA-DR+, p = 0.01) or unchanged (CD38+, p = 0.45). Most CVL cytokine and chemokine concentrations were not significantly different after implant placement except for a higher level of the soluble lymphocyte activation marker (sCD40L; p = 0.04) and lower levels of IL12p70 (p = 0.02) and G-CSF (p<0.001). In systemic blood, none of the changes noted in CVL after implant placement occurred except for decreases in the percentage CD4 T-cells expressing HLA-DR+ T cells (p = 0.006) and G-CSF (p = 0.02).

## Conclusions

Eng-Implant use was associated with a moderate increase in the availability of HIV target cells in the genital tract, however the percentage of these cells that were activated did not increase and there were minimal shifts in the overall immune environment. Given the mixed nature of these findings, it is unclear if these implant-induced changes alter HIV risk.

## Introduction

As approximately 40% of all pregnancies worldwide are unintended [1], prevention of unintended pregnancy is a public health priority. Over 200 million women worldwide use progestin-containing contraception (HC) either alone or in combination with estrogen to achieve their family planning goals [2]. Some research suggests that HC may contribute to the spread of HIV by increasing susceptibility to infection [3]. The greatest concern had been with Depot Medroxyprogesterone acetate (DMPA) where a meta-analysis of nine studies estimated a significant increase in HIV risk of 40% with DMPA compared to non-HC use [3]. Notably, these findings have not been consistently demonstrated by all studies, and all the studies were observational thus prone to potential confounding. Further, the findings have been challenged by the results from the Evidence for Contraceptive Options and HIV outcomes (ECHO) study [4], a large randomized trial that found no significant increase in HIV risk among users of DMPA compared to copper intrauterine device and levonorgestrel implant users. While these data are reassuring, gaps remain in our understanding of the relative HIV risk with contraceptive use compared to non-use and the impact of other progestin-containing contraceptive methods. Notably, data on the etonogestrel implant (Eng-Implant), are limited despite increasing rates of global use of these contraceptive implants.

There are several potential non-contraceptive effects from the use of potent steroid hormones. High levels of estrogen and progesterone during pregnancy are associated with a shift from a TH1 to a TH2-dominant immune profile, dampening the pro-inflammatory pathways, and increasing susceptibility to certain disease conditions (e.g., influenza, malaria, listeria) while reducing the severity of others (e.g., multiple sclerosis, rheumatoid arthritis) [5–9]. This natural phenomenon lends credence to the scientific premise of immune changes with hormone concentrations. HIV risk could thus be amplified by an increased representation of cells expressing HIV co-receptors within the female genital tract or trafficking of HIV target cells to the genital mucosa. CD4+ T-cells expressing the cell surface receptor C-C chemokine receptor type 5 (CCR5, the primary HIV co-receptor) are among the first cells to be infected and subsequently the virus can spread to regional lymph nodes [10]. Within the lower genital tract mucosa, the number and type of cellular targets, primarily CD4+ T-cells expressing CCR5, predict susceptibility to HIV infection [11, 12]. The functional properties exhibited by CD4+ T-cells influence susceptibility to HIV infection, specifically the expression of activation-associated molecules (markers such as HLADR, CD38) have been associated with increased risk of HIV acquisition [13–16].

Prior studies have evaluated the immune effects of DMPA on these cellular markers, with some, but not all, noting increases in key HIV target cell markers [17–23]. No studies have explored the effect of the Eng-Implant on these key cellular markers of HIV risk nor evaluated larger epidemiologic data to explore any cohorts to evaluate for an association between Eng-Implant use and HIV acquisition. Given this gap, we aimed to prospectively examine the effect of Eng-Implant initiation on the systemic and lower genital tract mucosal immune environment, with a focus on HIV target cells. The Eng-Implant is a long-acting highly effective progestin-only contraceptive method containing a 3rd generation progestin. Although Eng-Implant has less glucocorticoid activity compared to medroxyprogesterone in DMPA and less endogenous estrogen inhibition, we hypothesize that given the sustained progestin exposure over time, there will still be some immunologic changes within the genital tract to suggest increased susceptibility with use.

## Materials and methods

### Study population and recruitment

This was a prospective study to evaluate the effect of three months of Eng-Implant use on HIV target cells and inflammatory markers in the lower genital tract and systemic circulation. This manuscript is the first to evaluate one of the primary study objectives of a larger cohort study of three contraceptive methods registered at clinicaltrials.gov, NCT02357368. Women recruited into the larger cohort could initiate the Eng-Implant, the Levonorgestrel Intrauterine device or DMPA based on their preference. For this analysis, we will focus on the results from all individuals selecting the Eng-Implant, as the differences in baseline characteristics and relatively small sample size limit our power to make comparisons between methods. Women interested in initiating a new contraceptive method were recruited from the metro-Atlanta area via community-based postings or local referral from clinics. We enrolled eligible women between ages 18–45, who experienced normal menses (22–35 day intervals) for at least three cycles, had an intact uterus and cervix, and were HIV uninfected (determined by point-of-care rapid test using Ora-Quick®). Participants could not have used HC or copper intrauterine device (IUD) in the previous 6 months, had any signs of an STI on clinical examination at time of enrollment and needed to be medically eligible to initiate their selected contraceptive method (for this analysis Eng-Implant) based on CDC medical eligibility criteria for contraceptive use and clinical judgment. Approval for this study was obtained from the Emory IRB and Grady Research Oversight

Committee prior to study initiation. Written informed consent was obtained from all participants and all laboratory researchers and technicians were blinded to contraceptive exposure.

## Study procedures/clinical visits

The primary exposure of interest was the Eng-Implant (Nexplanon®, Merck & Co, Inc) [24]. We scheduled four study visits for each participant, two visits prior to contraceptive initiation and two visits approximately three months after Eng-Implant administration. Study visits were scheduled with the goal of pre-contraceptive sample collection at both the luteal (visit 1-target of 21 days after last menstrual period with window of 17 days to onset of next menses) and the follicular (visit 2-target three days after completion of menses, with window up to 14 days after onset of menses) phases of the menstrual cycle, based on a self-reporting of date of the last menstrual period. The Eng-Implant was placed at completion of visit 2. Post-contraceptive sampling collection occurred two weeks apart approximately three months after contraceptive initiation: visit 3 with target 12 weeks after (window 11–14 weeks after contraceptive initiation) and visit 4 with target 14 days after visit 3 (window of 12 to 16 days). The a priori goal was to compare the baseline (with follicular and luteal variation accounted for) with the post contraception (with 2 visits to account for variations over a 2 week period where endogenous hormonal changes may occur) results. We requested participants to abstain from vaginal intercourse for 24 hours prior to each visit to minimize the risk of contamination of genital tract samples by semen.

## Specimen collection

During a speculum examination, we collected a cervicovaginal swab for sexually transmitted infections (DrySwab™, Lakewood Biochemical Company). This was followed by a cervicovaginal lavage (CVL) collection with a lavage from the cervix, vaginal walls and posterior fornix with 10 ml of phosphate-buffered saline (PBS) for approximately 60 seconds as per the protocol described by the Microbicide Trials Network (https://vimeo.com/224957115/00cb72fed6) with details previously described [25]. To enhance cellular yield, CVL was performed twice. We collected blood in 8 mL sodium citrate-containing CPT tubes (BD Biosciences). CVL allows enrichment of target cells positioned at the apical lumen in proximity to exposure with a lower risk of tissue trauma from sampling that would cause bleeding and contamination of phenotyping. CVL or luminal cells are not imbedded within the tissue, persist within a harsh environment, and have a reduced cell yield compared with other sampling approaches, but CVL provides an accurate means of tissue resident phenotyping at the site of sexually transmitted exposure. In experiments where luminal T cells are analyzed separately from T cells embedded in the tissue, these two populations have been shown to be very similar phenotypically and functionally. [26–31] Microscopically, it has been shown that luminal T cells remain closely associated with the apical face of the epithelium. [32–36]Several studies have shown these luminal T cells are viable, capable of recognizing and responding to antigen, and play a critical role in immunity at mucosal sites[28, 35, 37–41]. Luminal T cells are sufficient to provide significant protection even when T cells located in the underlying tissues are not present [41], thus although you may not find a large number of T cells in CVL, these cells can be critical for barrier protection. With our methodology we have high viability of the cells (70–90%) and although cell count numbers for leukocytes are low, these counts are within the range of other sampling methodologies [25, 42].

## Covariate assessment

The following covariates were measured at each visit. 1) Semen presence in CVL: we detected semen presence using the Abacus ABAcard p30 test to detect prostate specific antigen (PSA). 2) Presence of sexually transmitted infections (STIs): we collected a cervicovaginal swab for STIs (DrySwab™, Lakewood Biochemical Company) at each visit prior to CVL from cervical os. DNA was extracted using the Qiagen DNA Mini Kit and used to amplify targets from *Neisseria gonorrhoeae*, *Chlamydia trachomatis*, *Mycoplasma genitalium*, *Trichomoniasis vaginalis*, and Herpes simplex virus types 1 and 2, using two real-time duplex PCR assays and Qiagen Rotor-Gene Q real-time PCR instrument. Qiagen Rotor-Gene Q Series software was used to analyze data. These multiplex PCR assays were performed in the Division of Sexually Transmitted Diseases Laboratory Reference and Research Branch at the US Centers for Disease Control and Prevention. 3) Bacterial Vaginosis (BV): we determined the presence of BV by Nugent score criteria [43] from gram stains prepared from CVL. Prior comparison data from our lab between CVL smears and swab among 37 sample pairs were highly correlated (r>0.88, p<0.0001) with categorical interpretations in agreement for all slides. Scores above six are considered consistent with BV. 4) Blood presence in the CVL: we defined blood presence qualitatively with a urine dipstick test detecting $> = 8000$ RBC/μl. This cut off was selected for inclusion of potential systemic blood contamination, however exploration and use of other cut-off values did not meaningfully alter our study findings.

## Immune marker assessment

Specimens were placed in a cooler with ice immediately after collection and transported to the Division of HIV/AIDS Prevention Laboratory Branch at Centers for Disease Control and Prevention (CDC) within four hours of collection for processing, cellular isolation and characterization. Blood was separated into plasma and peripheral blood mononuclear cells (PBMCs) by centrifugation in CPT tubes as instructed by manufacturer. After collecting blood plasma, PBMCs were collected from the CPT tube and washed with PBS prior to staining. CVL specimens were enriched for leukocytes using Percoll gradient centrifugation as previously described [25]. Plasma and CVL supernatant aliquots were stored at -80˚C until analysis. Cellular characterization was performed at that time on CVL leukocytes and PBMCs via flow cytometry. Viable leukocytes were distinguished using Zombie Fixable Viability Kits (Biolegend) then blocked for non-specific staining with anti-CD16/32 Fc (BioXcell).

The primary outcome of interest was the proportion of CD4 cells with CCR5 expression. Secondary outcomes evaluated were: 1) CD4/CD8 T-cell ratio to measure the changes in T-cell homeostasis, 2) the expression of activation markers CD38 and HLA-DR, peripheral tissue retention marker CD103[44] and trafficking marker CCR7 on CD4 T-cells or CD4 CCR5+ T-cells, 3) the differentiation of lymphocyte memory CD4 and CD8 T-cell phenotypes (Naïve T-cell ($T_{NA}$): CD45RA$^{hi}$ and CCR7$^{hi}$; central memory T-cells ($T_{CM}$): CD45RA$^{lo}$ and CCR7$^{hi}$; effector memory T-cells ($T_{EM}$): CD45RA$^{lo}$ and CCR7$^{lo}$; and effector memory RA expressing T-cells ($T_{EMRA}$): CD45RA$^{hi}$ and CCR7$^{lo}$). These outcomes are quantified for each individual at every time point.

Cells were stained with the following fluorochrome conjugated antibodies: CD3 (V450, UCHT1), CD4 (Alexa Fluor 700$^{®}$, RPA-T4), CD8 (BV510, RPA-T8), CCR7 (PE-CF594, 150503), CCR5 (PE, 3A9), CD103 (FITC, Ber-ACT8) (BD Biosciences), CD38 (PE/Cy7, HIT2), CD45 (BV650™, H130), CD45RA (BV605™, HI100), HLA-DR (BV785™, L243) (Biolegend). Stained samples were run on an LSRII flow cytometer and acquired using FACS DIVA software (BD Immunocytochemistry Systems, San Jose, CA, USA) and analyzed using FlowJo software (TreeStar, Inc.). Cellular measurements were analyzed as a percentage of CD4$^+$ T

cells, or CD4+ CCR5+ T cells expressing a given marker or combination of markers. For accurate measurement of CCR5 expression frequency on CD4 T cells, CCR5 gating was set against matched-naïve CD4 T cells from PBMCs as previously described [45].

Soluble immune mediators from the CVL supernatant and plasma were evaluated using Luminex technology with xPONENT software (Luminex Corporation) with all samples tested in duplicate on a 96 well plate containing seven standards, two quality controls and 39 samples using a customized multi-analyte panel (HCYTOMAG-60K-18 MILLIPLEX Human Cytokine panel, Millipore). The panel contained selected proinflammatory, inhibitory and chemotactic soluble cytokine and chemokines [IL-1b, IL-6, IL-12 (p70), IFN-a2, IFN-g, IL-1a, IL-17, IL-2, TNF-a, IL-4, GM-CSF, G-CSF sCD40, MIP-1a, MIP-1b, IP-10, IL-8, Fractalkine (CX3CL1)]. Using the sigmoid standard curve from the Millipore Analyst 5.1, a regression curve was extrapolated from the raw data individually for each cytokine. For samples below the level for quantification, we used half the lower limit of detection.

## Statistical methods

Visits were dichotomized into pre-implant use (visits 1 and 2) and post-implant use (visits 3 and 4). Any outcome (cytokine, cellular marker) value below the limit of detection was assigned a value of half of the lowest measured value for that outcome. Only samples with greater than 100 viableCD3+ cells extracted were included in the analyses, a similar approach to Lajoie et al [46]. To evaluate for potential associations that may be confounding our interpretation of study findings, we conducted separate logistic mixed models to assess the association of implant status (pre, post) with CVL visit characteristics (semen, STI, blood, BV). Models contain covariate of interest, a random intercept for subject, and variance components variance structure. This statistical approach was selected for evaluation of longitudinal data with repeated measurements of the same patient over time [47]. Separate generalized linear mixed models with a gamma distribution and log link were used to assess the association of each cytokine (IL-1b, IL-6, IL-12 (p70), IFN-a2, IFN-g, IL-1a, IL-17, IL-2, TNF-a, IL-4, GM-CSF, G-CSF sCD40), chemokine (MIP-1a, MIP-1b), chemotactic cytokine (IP-10, IL-8, Fractalkine) and cellular marker (CD4 CCR5, CD4 CD38, CD4 HLA-DR 2, CD4 CD103, CD4 CCR7) outcome with implant use (pre, post). Models included implant use, a random intercept for subject and variance components covariance structure. All models were stratified by tissue type (CVL, blood). CVL models additionally included presence of semen, presence of blood, STI, and BV status as covariates. Model-based estimates and 95% confidence intervals of estimated mean outcome level by implant use were back-transformed (exponentiated) to produce estimated arithmetic means on the original scale. Similarly, the estimated arithmetic mean ratio (AMR) and 95% confidence interval of post-implant use to pre-implant use was produced by exponentiating the coefficient for implant use. Linear mixed regression models were used to assess whether the distribution of lymphocyte memory cells into four mutually exclusive groups ($T_{NA}$, $T_{CM}$, $T_{EM}$, $T_{EMRA}$) varied by implant use. Models contained memory cell type, implant use, and memory cell type $^*$ implant use interaction term. The Type 3 F test of the interaction term is reported as well as model-based estimates and 95% confidence intervals for memory cell type and implant use. The models included robust variance estimates and compound symmetry covariance structure by subject grouped by memory cell type nested within implant use and were stratified by CD4 and CD8 T-cell phenotypes and tissue type. Model fit was assessed through residual plots. To reduce the potential impact from multiple comparisons on false discovery, we interpreted our results with an adjusted p-value using a Benjamini and Hochberg false discovery rate of 0.1 for each set of analyses within specimen

type and cytokine/cellular marker sets. Memory cell type analyses set $\alpha = 0.05$. Analyses were conducted in SAS v9.4.

## Results

Eighteen women enrolled in the study and completed both pre-contraceptive visits, 16 women completed visit 3 (88.9%) and 15 completed all 4 visits (83.3%). All pre-contraceptive luteal and follicular samples were collected during the appropriate windows described in the study methods, with Visit 3 and Visit 4 conducted at a median of 84 days (Q1: 83, Q3: 86.5 days) and 105 days (Q1: 98, Q3: 111) post-contraceptive initiation. Women were predominately African-American (83%), unmarried (83%), and young (median age 24 years) (Table 1). CVLs collected from 53 (79% of all visits) visits contained greater than 100 viable CD3+ T-cells and were subsequently included in this analysis. There were no associations between having fewer than 100 CD3+ T-cells on the analysis and visit number (data not shown). STIs were diagnosed by PCR at 28 (42%) visits, and BV diagnosed by Nugent score at 27 (42%) visits (Table 2). There were no significant differences in any of the visit level covariates between before and after the implant placement.

In the lower genital tract, we noted a significant increase in the proportion of CD4 cells expressing CCR5 after implant placement compared to measures taken prior to placement [AMR 1.56, 95% CI 1.09–2.24] (Table 3). Furthermore, there was a decrease in the CD4/CD8 ratio [AMR 0.70, 95% CI 0.53, 0.94], consistent with a significant increase in the proportion of CD8+ T-cells following Eng-Implant use [AMR 1.27, 95% CI: 1.06–1.53]. Eng-Implant use resulted in a decreased proportion of genital tract CD4+ T-cells expressing HLA-DR [AMR 0.58, 95% CI 0.38, 0.90] and CD4 CCR5+ T-cells expressing HLA-DR [AMR 0.54, 95% CI 0.34, 0.85]; however there were no significant changes in the expression of the activation marker CD38. Notably, all of these findings except for the decreased CD4 CCR5+ T-cells expressing HLA-DR remained significant after adjusting the alpha for multiple comparisons.

**Table 1. Cohort characteristics of n = 18 women enrolled in study.**

|  | n | % |
|---|---|---|
| **Age, years (median (Q1,Q3))** | 23.7 | (23.2, 30.2) |
| **Race** |  |  |
| African-American | 15 | (83.33) |
| Other | 3 | (16.67) |
| **Ethnicity: Hispanic** | 1 | (5.56) |
| **Marital Status** |  |  |
| Married/cohabitating | 3 | (16.67) |
| Single/divorced/widowed | 15 | (83.33) |
| **Education** |  |  |
| <High school diploma | 4 | (22.22) |
| High school diploma/GED | 5 | (27.78) |
| Some college | 6 | (33.33) |
| Associate's degree/Technical certification | 1 | (5.56) |
| Bachelor's degree | 2 | (11.11) |
| **Annual Income** |  |  |
| <$10,000 | 7 | (38.89) |
| $10,000-$24,999 | 5 | (27.78) |
| $25,000-$50,000 | 4 | (22.22) |
| Don't know/refuse | 2 | (11.11) |

**Table 2. Time-varying characteristics of CVL specimens by study visit.**

| Characteristic | Visit 1 (n = 18) | | Visit 2 (n = 18) | | Visit 3 (n = 16) | | Visit 4 (n = 15) | |
|---|---|---|---|---|---|---|---|---|
| | n | % | n | % | n | % | n | % |
| **Semen** | 4 | (22.2) | 5 | (27.8) | 2 | (12.5) | 3 | (20.0) |
| **STI** | 5 | (27.8) | 7 | (38.9) | 9 | (56.3) | 7 | (46.7) |
| **BV** | 6 | (33.3) | 8 | (47.1) | 7 | (50.0) | 6 | (40.0) |
| **Blood** | 1 | (5.6) | 4 | (22.2) | 2 | (12.5) | 1 | (6.7) |
| **Viable CD3 lymphocyte count <100** | 6 | (33.3) | 3 | (16.7) | 2 | (12.5) | 3 | (20.0) |
| | Median | IQR | Median | IQR | Median | IQR | Median | IQR |
| viable CD3 lymphocyte number | 548.5 | 245, 2769 | 1118 | 185, 2330 | 1629 | 844, 4121 | 1036.5 | 423, 3288.5 |
| CD4 lymphocyte number* | 328 | 140, 1340 | 494 | 128, 998 | 769 | 346, 1990 | 574 | 76, 1418 |

IQR = Interquartile range,

* among samples with viable CD3 lymphocyte count $\geq$ 100.

Additionally, Eng-Implant use significantly changed the distribution of T-cell subtypes among both the CD4 (p = 0.004, Fig 1A) and CD8 T-cells (p = 0.023, Fig 1B), with an observed shift away from effector memory subtype.

Among the PBMCs, there were minimal changes in the distribution of T-cell phenotypes, with no noted changes in T-cell ratios or CCR5 co-receptor expression (Table 4). There was a significant increase in CCR7 expression on CCR5+ CD4 T-cells [AMR 1.30, 95% CI 1.08, 1.57] and decreased HLA-DR on CD4 T-cells [AMR 0.81, 95% CI 0.70, 0.90]. These findings remained significant with the adjusted alpha value. There was a significant difference in distribution of memory cell phenotype among the CD4 T-cells (p = 0.014) with an observed shift towards more naïve cells and reduced effector memory cells (Fig 2A and 2B).

**Table 3. Estimated cellular marker levels in the CVL for implant users, adjusting for repeated measures and covariates.**

| | Pre-Implant | Post-Implant | | Arithmetic Mean Ratio Post-Implant/ Pre-Implant |
|---|---|---|---|---|
| Cellular Marker | Estimate* (95%CI) | Estimate* (95% CI) | p-value | (95% CI) |
| **% of CD3 + T-cells expressing:** | | | | |
| CD4+ | 51.07 (41.45, 62.93) | 46.92 (38.41, 57.32) | 0.373 | 0.92 (0.76, 1.11) |
| CD8+ | 21.60 (16.64, 28.03) | 27.48 (21.28, 35.49) | **0.013** | 1.27 (1.06, 1.53) |
| CD4/CD8 ratio | 2.60 (1.81, 3.74) | 1.83 (1.29, 2.61) | **0.019** | 0.70 (0.53, 0.94) |
| **% of CD4+ T-cells expressing:** | | | | |
| CCR5+ | 17.76 (10.64, 29.65) | 27.75 (16.69, 46.14) | **0.017** | 1.56 (1.09, 2.24) |
| CD38+ | 32.37 (24.56, 42.66) | 40.26 (31.06, 52.19) | 0.111 | 1.24 (0.95, 1.63) |
| HLA-DR+ | 25.69 (15.21, 43.38) | 15.01 (8.96, 25.13) | **0.018** | 0.58 (0.38, 0.90) |
| CD103+ | 6.58 (3.09, 14.02) | 11.32 (5.81, 22.08) | 0.084 | 1.72 (0.92, 3.21) |
| **% of CD4+ CCR5+ T-cells expressing:** | | | | |
| CCR7+ | 32.37 (22.02, 47.57) | 38.46 (27.11, 54.56) | 0.290 | 1.19 (0.85, 1.65) |
| CD38+ | 65.01 (51.84, 81.52) | 60.12 (49.54, 72.95) | 0.453 | 0.92 (0.75, 1.14) |
| HLA-DR+ | 50.98 (31.32, 82.98) | 27.45 (17.98, 41.89) | **0.009** | 0.54 (0.34, 0.85) |

Generalized linear mixed model with a random intercept for participant, variance components covariance structure, gamma distribution, log link Restricted to CD3 count>100

* Back-transformed estimate (arithmetic mean)

** P-value is for adjusted model. Bold indicates significant after Benjamini–Hochberg correction

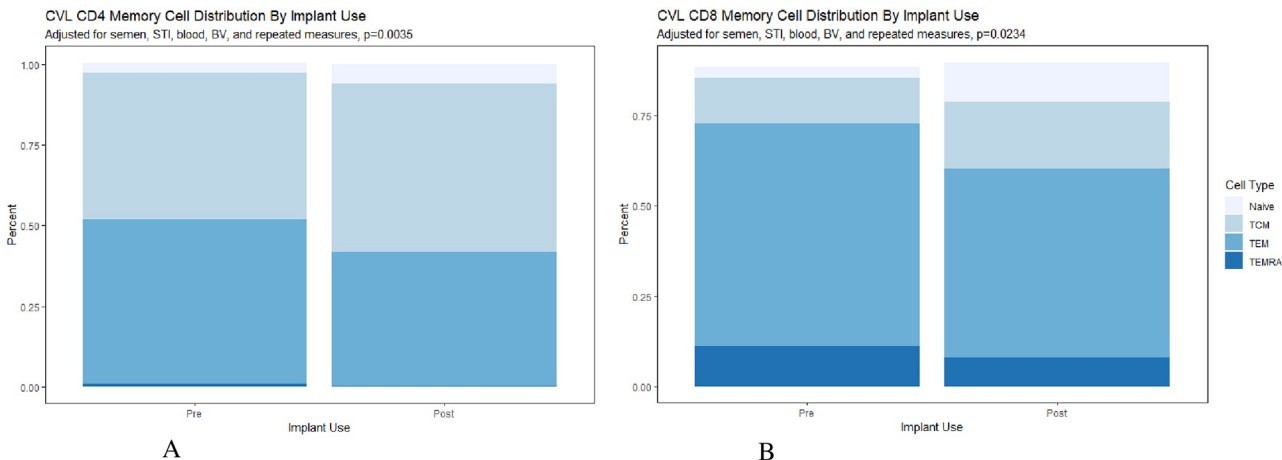

**Fig 1.** a. CVL CD4 memory cell distribution before and after implant use, adjusting for time-varying semen presence by PSA, sexually transmitted infections, bacterial vaginosis, presence of blood in the sample and repeated measures, p = 0.004. b. CVL CD8 memory cell distribution before and after implant use, adjusting for time-varying semen presence by PSA, sexually transmitted infections, bacterial vaginosis, presence of blood in the sample and repeated measures p = 0.023. Memory cell phenotypes included Naïve T-cell (CD45RA$^{hi}$ and CCR7$^{hi}$); Tcm = central memory T-cells (CD45RA$^{lo}$ and CCR7$^{hi}$; Tem = effector memory T-cells (CD45RA$^{lo}$ and CCR7$^{lo}$) and T$_{EMRA}$ = effector memory RA expressing T-cells (CD45RA$^{hi}$ and CCR7$^{lo}$).

Overall, lower genital tract cytokine expression was similar before and after implant initiation (Fig 3, S1 Appendix), with significant reductions only noted for GSCF [AMR 0.54, 95% CI 0.39–0.74, p = 0.0004] and IL12p70 [AMR 0.69, 95% CI 0.51–0.94, p = 0.0212] and a significant increase in sCD40L [AMR 1.46 95% CI 1.02, 2.08, p = 0.0380]. Among these, only GCSF remained significant after adjusting the alpha for the multiple comparisons. Similarly minimal changes in plasma cytokine concentrations were observed following Eng-Implant use, with

**Table 4. Estimated cellular marker levels in PBMC for implant users, adjusting for repeated measures.**

| Cellular Marker | Pre-Implant Estimate* (95% CI) | Post-Implant Estimate* (95% CI) | p-value** | Arithmetic Mean Ratio Post-Implant/ Pre-Implant (95% CI) |
|---|---|---|---|---|
| **% of CD3 + T-cells expressing:** | | | | |
| CD4+ | 66.48 (62.58, 70.61) | 66.24 (62.25, 70.49) | 0.857 | 1.00 (0.96, 1.04) |
| CD8+ | 25.10 (21.75, 28.97) | 25.69 (22.14, 29.80) | 0.662 | 1.02 (0.92, 1.14) |
| CD4/CD8 ratio | 2.80 (2.22, 3.54) | 2.77 (2.17, 3.55) | 0.929 | 0.99 (0.81, 1.21) |
| **% of CD4+ T-cells expressing:** | | | | |
| CCR5+ | 3.46 (2.64, 4.53) | 3.97 (2.99, 5.27) | 0.276 | 1.15 (0.89, 1.48) |
| CD38+ | 15.76 (12.23, 20.32) | 15.41 (11.85, 20.04) | 0.800 | 0.98 (0.82, 1.17) |
| HLA-DR+ | 2.78 (2.32, 3.33) | 2.26 (1.87, 2.73) | **0.006** | 0.81 (0.70, 0.94) |
| CD103+ | 0.14 (0.07, 0.28) | 0.12 (0.05, 0.24) | 0.351 | 0.81 (0.52, 1.28) |
| **% of CD4+ CCR5+ T-cells expressing:** | | | | |
| CCR7+ | 25.81 (20.96, 31.77) | 33.65 (27.15, 41.70) | **0.007** | 1.30 (1.08, 1.57) |
| CD38+ | 24.58 (20.21, 29.88) | 27.02 (22.08, 33.05) | 0.290 | 1.10 (0.92, 1.31) |
| HLA-DR+ | 22.22 (19.52, 25.29) | 21.08 (18.41, 24.14) | 0.496 | 0.95 (0.81, 1.11) |

Generalized linear mixed model controlling for time-varying semen presence by PSA, sexually transmitted infections, bacterial vaginosis, presence of blood in the sample and repeated measures with a random intercept for participant, variance components covariance structure, gamma distribution, log link. Restricted to CD3 count>100

* Back-transformed estimate (arithmetic mean)

** P-value is for adjusted model. Bold indicates significant after Benjamini–Hochberg correction

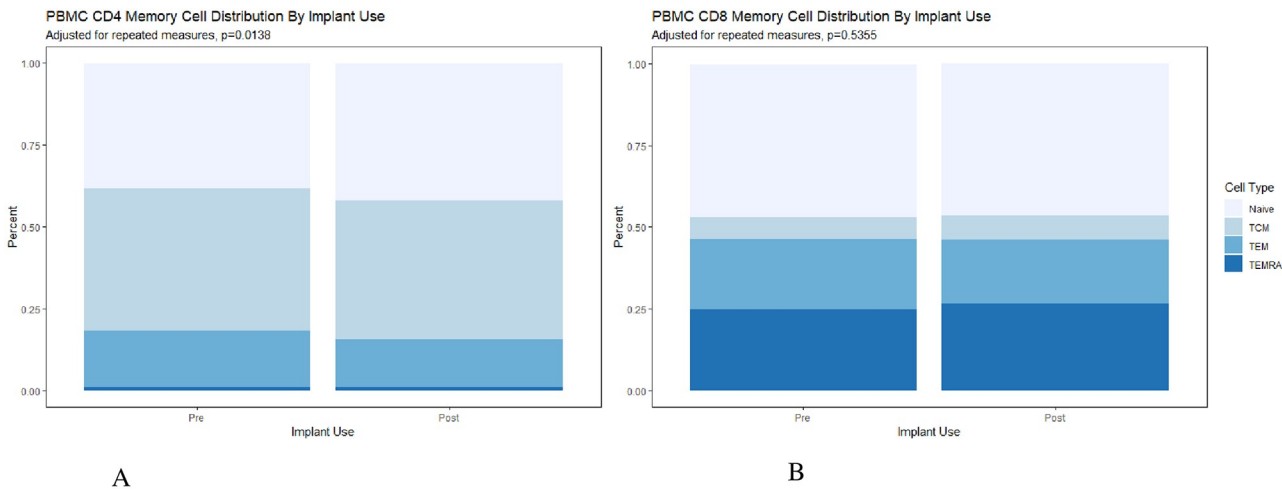

**Fig 2.** a. PBMC CD4 memory cell distribution before and after implant use, adjusting for repeated measures, p = 0.014. b. PBMC CD8 memory cell distribution before and after implant use, adjusting for repeated measures, p = 0.536. Memory cell phenotypes included Naïve T-cell (CD45RA$^{hi}$ and CCR7$^{hi}$); Tcm = central memory T-cells (CD45RA$^{lo}$ and CCR7$^{hi}$; Tem = effector memory T-cells (CD45RA$^{lo}$ and CCR7$^{lo}$) and T$_{EMRA}$ = effector memory RA expressing T-cells (CD45RA$^{hi}$ and CCR7$^{lo}$).

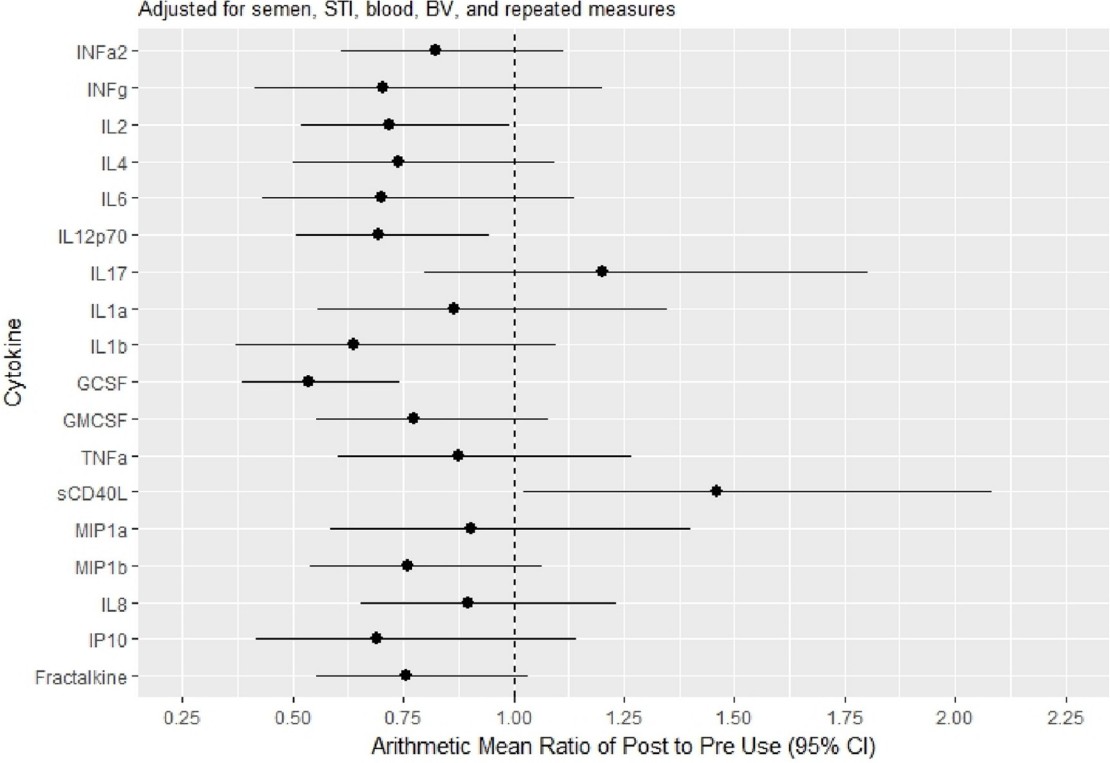

**Fig 3. Forest plot of arithmetic mean ratio of cytokine levels post-implant compared to pre-implant use with 95% confidence intervals, adjusting for time-varying semen presence by PSA, sexually transmitted infections, bacterial vaginosis, presence of blood in the sample and repeated measures.**

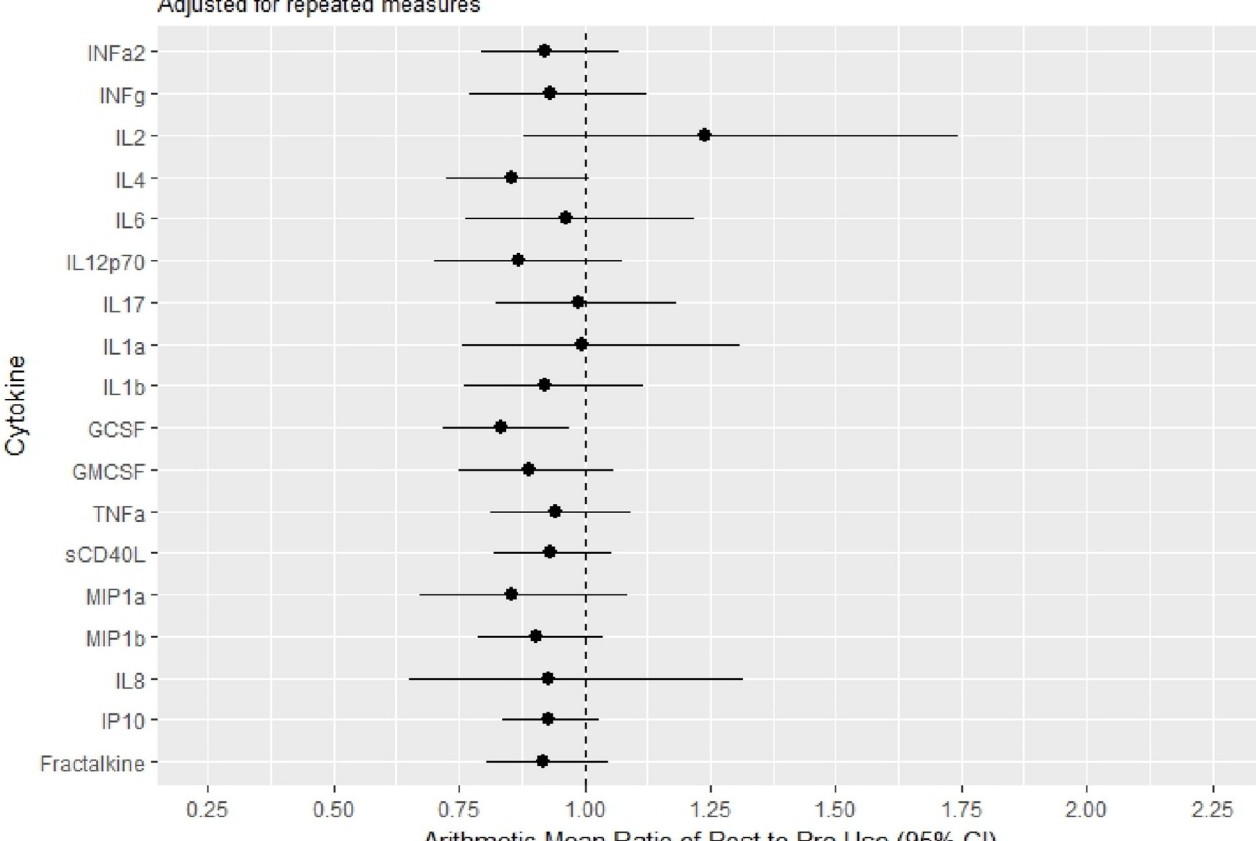

**Fig 4. Forest plot of arithmetic mean ratio of cytokine levels post-implant compared to pre-implant use with 95% confidence intervals, adjusting for repeated measures.**

only a significant reduction in GCSF [AMR 0.84, 95% CI 0.72, 0.97, p = 0.0207] (Fig 4, S2 Appendix), however this was no longer significant after adjusting alpha for multiple comparisons.

## Discussion

The results of our study highlight few changes in the lower genital tract inflammatory environment following Eng-Implant initiation. While several studies have evaluated genital tract immune changes after use of other hormonal contraceptive methods [17–23, 48–61], to our knowledge, no published study has evaluated Eng-Implant. We report an increase in the proportion of CD4 T-cells expressing the co-receptor CCR5 at the genital mucosa with Eng-Implant use that could be associated with increased risk of HIV infection; however, not all our findings relate a clear picture of increased susceptibility. For example, implant placement reduced the frequency of the activation marker HLA-DR, but not CD38, among CD4 T-cells. The clinical significance of these findings on HIV acquisition is unclear. Further, while there were minimal changes in soluble immune markers in the lower genital tract, these changes suggest a slight increase in local immune suppression with reduced concentrations of proinflammatory cytokines, a finding similarly noted with DMPA [54]. As we observed some shifts in T-cell populations in the genital tract

associated with Eng-Implant use, we cannot eliminate the potential that Eng-Implant has an effect on HIV acquisition. This finding is important when interpreting the results of the recently conducted ECHO study [4], as they did not find a significant difference in HIV acquisition between DMPA users and users of the copper intrauterine device and the levonorgestrel implant. The ECHO results are encouraging that DMPA did not differ from these other methods in relation to HIV risk. The ECHO study was powered to detect a clinically significant increased risk of 50% and conclusions regarding other methods cannot be made. Furthermore, while we evaluated the etonogestrel and not levonorgestrel implant, we find some changes in immunologic markers with unclear impact on susceptibility. While small changes in individual risk with a contraceptive method use should not alter eligibility for use [62] of a contraceptive method, with increasing global utilization of many of the longer-acting contraceptive methods, it is important for research to identify even subtle differences that may influence counselling for high-risk individuals and have public health importance.

The increased expression of CCR5 at the genital mucosa may reflect infiltration of T-cells or a direct effect of the implant on the expression of CCR5 [63]. The increased expression of CD103, coupled with the shift from the canonical effector memory phenotype towards a central or migratory memory subtype, suggests that infiltration and retention are drivers of this shift [25]. The CD4/CD8 T-cell ratio further supports that the Eng-Implant use is influencing the trafficking patterns of immune cells. The increased frequency of CD8 T-cells at the genital mucosa is provocative and clue potential alterations in local inflammation. Prior research suggests that effector CD8 T-cells cannot enter into the vagina without CD4-T cell permission in the form of activation-associated cytokines [33]. While our cytokine findings do not fully support this finding, it is possible that soluble cytokine measurements may not detect this mechanism.

A decrease in GCSF was noted in the genital tract after implant initiation. A similar reduction in GCSF was also observed in plasma (although not significant after adjusting for multiple comparisons). This finding of a small yet significant reduction in GCSF may signify an alternative pathway associated with altered HIV susceptibility through damaged mucosa. Granulocyte colony-stimulating factor (GCSF) may induce an inflammatory reaction enhancing neutrophil function. With receptors on granulosa cells, GCSF has been implicated in ovulation and thus could be downregulated in the setting of ovulation inhibition associated with implant use [64]. GCSF is also associated with wound healing and has been associated with faster healing from genital ulcerations [65], GCSF stimulates the proliferation and differentiation of cells that participate in acute and chronic inflammation and immune responses including mature leukocytes, macrophages, and dendritic cells [66]. This potential mechanism of altered immune response should be further explored to determine if clinically significant.

The mechanism by which progestin contraceptives may be influencing immune expression in the genital tract is not fully elucidated. Progestins may act via alteration of gene expression after binding to and activating intracellular steroid receptors [63], which vary based on different tissue cell types. Gonadal hormones can regulate the expression of numerous genes involved in multiple cellular functions [67, 68] with the effects modified by cell type, presence of other hormones and transcription factors, and their binding potential for progestins by other steroid receptors besides the progesterone receptor can result in agonist or antagonist activity. Further, biological effect can vary based upon the dose of progestin. High progestin levels can cause thickening of cervical mucus that creates a barrier to sperm assent, suppress ovulation and alter the endometrial lining. Progestin-containing contraceptive methods can differ by their mode of delivery, length of effectiveness, global availability, degree of endogenous hormone and ovulation inhibition and type of progestin they contain with varying degrees of estrogenic, androgenic, anti-androgenic, glucocorticoid and anti-mineralocorticoid

activity [69, 70]. For example, medroxyprogesterone (MPA), a synthetic progestin in DMPA, has potent glucocorticoid (GC) activity compared with weaker GC activity for ENG, where levoneorgestrel (LNG) has no GC activity. Given the differences among different contraceptive methods, it is important to understand the immunologic effect of varying types of contraception to understand their potential and relative impact on reproductive health and immunity. Notably, even among individuals using the same contraceptive, the serum progestin concentrations can vary widely and these differences may influence the effect of the particular contraceptive [63]. The Eng-Implant users may have variability in serum concentrations with implant use as well as tissue level exposure and tissue responsiveness via steroid receptors. Understanding the individual level factors that influence both systemic hormonal concentrations and mucosal level response to the hormone is critical to provide guidance for individual level counseling.

While other studies have not evaluated the Eng-Implant, prior studies evaluating the effects of DMPA have conflicting results [17–22]. For example, while two studies did not see changes in the vaginal HIV target cells with DMPA use [18, 23] one other found significantly higher frequencies of CCR5+ CD4+ T-cells (relative risk: 3.92) compared to non-users [22]. A recent cross sectional analysis comparing 15 DMPA users to 20 non-hormonal contraceptive users found higher levels of activated T-cells and a higher proportion of CD4+CCR5+ T-cells among DMPA users on tissue biopsy samples, however this increase was not noted among the cervical mononuclear cells obtained via cytobrush and cervical spatula [46]. Some of the inconsistencies may be related to the differences in study population, study methodology (cross sectional versus longitudinal), timing of sample collection in relation to luteal or follicular phases or timing in relation to hormonal contraception, sample collection approach or laboratory methodology. Importantly, vaginal immune parameters are influenced by many factors and quite variable within and between individuals. This variability may account for some of the discrepancies in DMPA studies and highlight the need to interpret the results of our study in the context of future research among different study populations and exploring individual level factors that may account for variable responses.

A strength of our study design is that we captured two time points over the course of four weeks both before and after implant placement to capture the overall environment given changes over a cycle with endogenous hormonal exposure. Given our small sample size, these results should be interpreted cautiously. While our sample size limits our power to evaluate subtle immunologic changes, the changes that we do identify highlight the need for larger, more robust studies to determine if these changes influence a woman's susceptibility to HIV infection. The longitudinal nature of this study allows us to control for measured and unmeasured biases that occur in cross sectional studies that are the predominant study type in the field. Although we excluded women from participation with clinical evidence of any infection at baseline, several women had asymptomatic infections diagnosed or acquired infections over the course of the study. As individuals with sexually transmitted infections, bacterial vaginosis, and recent semen exposure, factors independently known to alter HIV susceptibility, were not excluded from this analysis, but rather the time-varying presence of these exposures were controlled for in our final models, we feel these findings are likely more representative of real-world findings. Although BV, STIs and semen may modify HIV susceptibility, larger studies are needed for adequate power to analyze the potential effect modification of these risk factors. Although heterogeneity in the endogenous hormonal response to the contraceptive is possible and we did not measure and control for endogenous hormonal levels, we selected to include 2 time points post initiation to help control for some of that variability. As there are known variations in local immune factors with these infections, the inclusion of these women may have contributed to reduced power for detecting a difference in some study outcomes. As women

are self-selected, individual differences that could underlie differential responses to contraceptive exposure may limit the generalizability of our results. Importantly, given the wide range of variability in the number of cells from the CVL that are collected, we are evaluating the proportion of cells expressing different cellular markers and not the number of total cells present that express these markers. Lastly, as we are also reporting on markers of HIV susceptibility, any extrapolation to qualify the degree that these factors may alter true susceptibility is limited.

There are multiple benefits of contraceptives beyond fertility-control including reduced abortion, maternal and neonatal morbidity and HIV perinatal transmission. Our findings relating Eng-Implant with HIV susceptibility markers are subtle with unclear clinical impact, and consistent with the results ECHO trial findings. Informed decision-making must include information about the superior typical-use effectiveness of long-acting reversible contraceptive methods, such as the Eng-Implant. Additionally, informed consent requires that we share information on the lack of clear evidence on increased risk of HIV susceptibility with all hormonal contraceptive methods with the promotion of dual method use with condoms.

## Supporting information

**S1 Appendix.**
(DOCX)

**S2 Appendix.**
(DOCX)

## Acknowledgments

Thank you to Tammy Evans-Strickfaden for assistance in the development of laboratory procedures and processes leveraged for this evaluation and Cheng Chen and Kai-Hua Chi for conducting the multiplex PCR assessment.

**CDC Disclaimer:** The findings and conclusions in this report are those of the authors and do not necessarily represent the official position of the Centers for Disease Control and Prevention.

## Author Contributions

**Conceptualization:** Lisa B. Haddad, Alison Swaims-Kohlmeier, Richard E. Haaland, Anandi N. Sheth, Sharon L. Achilles, Clyde E. Hart, Igho Ofotokun.

**Data curation:** Lisa B. Haddad, Alison Swaims-Kohlmeier, Nakita L. Brown, Anandi N. Sheth, Davis Lupo, Igho Ofotokun.

**Formal analysis:** Lisa B. Haddad, Alison Swaims-Kohlmeier, C. Christina Mehta, Hsin Chien.

**Funding acquisition:** Lisa B. Haddad.

**Investigation:** Lisa B. Haddad, Alison Swaims-Kohlmeier, Kehmia Titanji, Clyde E. Hart, Igho Ofotokun.

**Methodology:** Lisa B. Haddad, Alison Swaims-Kohlmeier, Richard E. Haaland, Hsin Chien.

**Project administration:** Lisa B. Haddad, Nakita L. Brown.

**Writing – original draft:** Lisa B. Haddad, Alison Swaims-Kohlmeier, C. Christina Mehta, Richard E. Haaland, Anandi N. Sheth.

**Writing – review & editing:** Lisa B. Haddad, Alison Swaims-Kohlmeier, C. Christina Mehta, Richard E. Haaland, Nakita L. Brown, Anandi N. Sheth, Hsin Chien, Kehmia Titanji, Sharon L. Achilles, Davis Lupo, Clyde E. Hart, Igho Ofotokun.

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
