## [Decision Letter · Decision Letter 0]

29 Oct 2019

PONE-D-19-23634

Impact of etonogestrel implant use on T-cell and cytokine profiles in the female genital tract and blood

PLOS ONE

Dear Dr Haddad,

Thank you for submitting your manuscript to PLOS ONE. After careful consideration, we feel that it has merit but does not fully meet PLOS ONE’s publication criteria as it currently stands. Therefore, we invite you to submit a revised version of the manuscript that addresses the points raised during the review process.

ACADEMIC EDITOR:

We would appreciate receiving your revised manuscript by Dec 13 2019 11:59PM. To enhance the reproducibility of your results, we recommend that if applicable you deposit your laboratory protocols in protocols.io, where a protocol can be assigned its own identifier (DOI) such that it can be cited independently in the future. For instructions see: http://journals.plos.org/plosone/s/submission-guidelines#loc-laboratory-protocols

We look forward to receiving your revised manuscript.

Kind regards,

Manish Sagar, MD

Academic Editor

PLOS ONE

Journal Requirements:

Additional Editor Comments:

Please see comments below from the reviewers.

As an editor, I am a bit worried about using cells from vaginal lavage. Often cell numbers are low especially live cells. This can make it difficult to ascertain phenotypes with certainty

Reviewers' comments:

Reviewer's Responses to Questions

**Comments to the Author**

1. Is the manuscript technically sound, and do the data support the conclusions?

Reviewer #1: Yes

Reviewer #2: Partly

Reviewer #3: Partly

2. Has the statistical analysis been performed appropriately and rigorously? 

Reviewer #1: Yes

Reviewer #2: Yes

Reviewer #3: Yes

3. Have the authors made all data underlying the findings in their manuscript fully available?

Reviewer #1: No

Reviewer #2: No

Reviewer #3: Yes

4. Is the manuscript presented in an intelligible fashion and written in standard English?

Reviewer #1: Yes

Reviewer #2: Yes

Reviewer #3: Yes

5. Review Comments to the Author

Reviewer #1: This manuscript reports the results from a prospective study investing the potential risk of etonogestrel implant use on change of T-cell and cytokine profiles in the female genital tract and blood. I have below comments or questions.

In Statistical Methods, for analysis of lymphocyte memory cells with four group types, please make it clear if each subject would have certain % of each of the 4 types of memory cells at each time point or each subject would only have one type of the 4 types of memory cells at each time point.

Per the recent publication (Nature Communications 10, Article number: 3753 (2019)), semen may have effect to alter the risk of HIV acquisition. It would be informative in this study to add some analysis to investigating the potential interaction effect between semen presence/absence and etonogestrel implant use.

In Tables 2 and 3, are the p-values the FDR adjusted p-values or the original p-values?

Please provide more information in a table to show the % and N at each time points for semen presence, STI, blood and BV.

Reviewer #2: Please justify the use of vaginal lavage samples for cell phenotyping, and provide cell numbers and cytokine concentrations in addition to percent change.

Explain how your data relates to data from DMPA studies; vaginal immune parameters are highly variable and could account for differences between studies.

Reword conclusions to remove the implication that data from your study suggests that HIV susceptibility is enhanced by Eng treatment.

Reviewer #3: This article describes the results of a prospective, observational study of changes in HIV target cells and cytokine concentrations associated with use of the ENG contraceptive implant. Few data have been published on the association of ENG implant use and HIV risk to date. With implant use increasing rapidly in areas of high HIV prevalence (most notably in sub-Saharan Africa), the study addresses an important gap in the scientific literature on ENG implant use and HIV susceptibility. In their analysis, the authors evaluated changes in immunological markers in cervicovaginal fluid and blood PBMCs and identified associations between ENG implant use and increases in the proportion of CD4+ cells expressing CCR5 but decreases in the CD4:CD8 T cell ratio in the genital tract only. Results of analyses assessing markers of T cell activation were inconsistent in both tissue types and few changes in cytokine concentrations were identified after ENG implant initiation. This study’s strengths lie in its prospective design which accounts for potential variation in immunological markers at different stages of the menstrual cycle and adequate ascertainment and control of important covariates (STIs, BV). The study is limited by its small sample size as only 15 participants had data from all four pre/post implant study visits. Given this limitation and the fact that these data are drawn from an observational design, which could further be confounded by unmeasured factors, results should be interpreted carefully.

Major comments

• Given the observational nature of this study, the authors should avoid causal language in the interpretation of their results, such as how implant use had an “effect on”, “results in” or “led to” changes in immunological markers. It’s highly recommended that this language be amended throughout the paper.

• Additionally, caution should be taken when interpreting changes in immunological markers as changes in HIV risk. Based on the final two sentences in the first paragraph of the discussion, it sounds as if the authors are suggesting that counseling around implant use should be altered based on the results of this study. This may be a step too far without evidence linking implant use with HIV acquisition. If anything, these results suggest that implant use may alter immunologic factors that have been shown to be associated with HIV risk, and highlight the need for further evidence. Please consider revising.

• The introduction could benefit from a clearer statement of the authors’ hypotheses and the biological or epidemiological evidence to support these hypotheses. Is there reason to believe from these studies that ENG exposure would increase HIV risk or alter immunity in the genital tract? Why is this research question worth pursuing aside from the current lack of evidence?

• Clarification of the sampling is needed in the methods. Were all implant users from the parent study included? If not, how were ENG implant users sampled and is there reason to believe there is selection bias from the sampling?

• Please provide rationale for the decision to impute the lowest measured value for BLQ results rather than using an approach that is independent of measured values (e.g half of the LLOQ). Would this not bias your results?

Minor comments

• In the introduction, it is worth noting that all of the studies that were conducted prior to ECHO were observational

• In the Results, consider adding the median/range of number of days from LMP that samples were collected for each pre-implant time point, and median/range number of days from implant insertion for the two post-implant time points. It is unclear whether you were able to collect the samples within the defined target luteal/follicular periods.

• What was the purpose of the models assessing the association between implant status with the CVL visit characteristics? Please clarify.

• In the models assessing the distribution of lymphocyte memory cells, a linear model seems inappropriate with a categorical outcome. Can you clarify what your dependent variable was in these models or further explain your rationale for the linear model?

• Could you please elaborate on the types of STIs diagnosed in the first paragraph of the Results?

• Did occurrence of detecting <100 CD3+ T cells vary significantly by the four visit types? If so, explain how this might influence your results.

• It is interesting that only a decrease in GCSF was noted in the genital tract after implant initiation. A similar reduction in GCSF was also observed in plasma (although not significant after adjusting for multiple comparisons). Consider discussing the role of this cytokine in the discussion and how it may relate to ENG exposure from a biological perspective.

• In the third paragraph of the discussion, the sentence beginning “Progestins can regulate the expression of numerous genes involved in multiple cellular functions…” should have a citation.

• Author group for citation 4 (ECHO paper) needs to be corrected

• It would be helpful to spell out the memory cell classifications in Figures 1 and 2 either in the legend or the description

6. PLOS authors have the option to publish the peer review history of their article (what does this mean?). If published, this will include your full peer review and any attached files.

Reviewer #1: No

Reviewer #2: No

Reviewer #3: No

---

## [Author Response · Author response to Decision Letter 0]

31 Jan 2020

REVIEWER COMMENTS:

Additional Editor Comments:

Please see comments below from the reviewers.

As an editor, I am a bit worried about using cells from vaginal lavage. Often cell numbers are low especially live cells. This can make it difficult to ascertain phenotypes with certainty

While we recognize your concern regarding CVL, we and several others have successfully characterized the FGT from CVL. CVL allows enrichment of target cells positioned at the apical lumen in proximity to exposure with a lower risk of tissue trauma from sampling that would cause bleeding and contamination of phenotyping. These CVL enriched lymphocytes are phenotypically and functionally shown to be comparable to those resident at the underlying tissue. With our methodology we have high viability of the cells (70-90%) and although cell count numbers for leukocytes are low, these counts are within the range of other sampling methodologies. We have added clarification to highlight this in the methods section. (Page 7, 3rd paragraph)

Reviewer #1: This manuscript reports the results from a prospective study investing the potential risk of etonogestrel implant use on change of T-cell and cytokine profiles in the female genital tract and blood. I have below comments or questions.

In Statistical Methods, for analysis of lymphocyte memory cells with four group types, please make it clear if each subject would have certain % of each of the 4 types of memory cells at each time point or each subject would only have one type of the 4 types of memory cells at each time point.

At each time point, for each individual, the cells are classified into 4 sub-types based on their marker expression. This clarification has been added to the methods section. (Page 9, 5th paragraph)

Per the recent publication (Nature Communications 10, Article number: 3753 (2019)), semen may have effect to alter the risk of HIV acquisition. It would be informative in this study to add some analysis to investigating the potential interaction effect between semen presence/absence and etonogestrel implant use.

We recognize that semen, similar to STIs and BV, may modify the impact of the implant on HIV susceptibility, however our study was not powered to be able to evaluate effect modification. We altered our discussion as follows and added this citation to the paper:

“As individuals with sexually transmitted infections, bacterial vaginosis, and recent semen exposure, factors independently known to alter HIV susceptibility, were not excluded from this analysis, but rather the time-varying presence of these exposures were controlled for in our final models, we feel these findings are likely more representative of real-world findings. Although BV, STIs and semen may modify HIV susceptibility, larger studies are needed for adequate power to analyze the potential effect modification of these risk factors.” )(Page 16, 6th paragraph)

In Tables 2 and 3, are the p-values the FDR adjusted p-values or the original p-values?

These are the adjusted p-values. This detail has been highlighted. (page 23 and 25, table 2 and 3)

Please provide more information in a table to show the % and N at each time points for semen presence, STI, blood and BV.

We have added Table 2 with the time varying characteristics at each visit. (Page 22, Table 2)

Reviewer #2: Please justify the use of vaginal lavage samples for cell phenotyping, and provide cell numbers and cytokine concentrations in addition to percent change.

As mentioned above, CVL allows enrichment of lymphocytes positioned at the apical lumen in proximity to exposure with a lower risk of tissue trauma from sampling that would cause bleeding and contamination of phenotyping. These CVL enriched lymphocytes are phenotypically and functionally shown to be comparable to those resident at the underlying tissue. The cellular yield for the CVL procedure can be variable and thus we rely on the percentages to aid in quantifying the comparable phenotype of the cellular populations. We have added detail in our methods section to highlight why we selected CVL for our analyses. 9table 7, 3rd paragraph)

Further we have added in median and IQR cell counts to table 2 and added supplementary tables that include the cytokine concentrations. We believe however that the proportion expressing these markers are more appropriate method of evaluation and have maintained our analyses as such. This approach is consistent with other studies utilizing CVL given variability in the number of cells retrieved with this sampling approach. In our discussion, we comment on this limitation. We do however believe that proportion of cells expressing different phenotypic markers highlight the characteristic quality of the immune response.

Explain how your data relates to data from DMPA studies; vaginal immune parameters are highly variable and could account for differences between studies.

Thank you for this comment. We have added the following to our discussion to address this point:

“Importantly, vaginal immune parameters are influences by many factors and quite variable within and between individuals. This variability may account for some of the discrepancies in DMPA studies and highlight the need to interpret the results of our study in the context of future research among different study populations and exploring individual level factors that may account for variable responses.” (Page 16, 5th paragraph)

Reword conclusions to remove the implication that data from your study suggests that HIV susceptibility is enhanced by Eng treatment.

We agree that the conclusion of enhanced susceptibility with Eng treatment should be restated. Our conclusions were written to highlight that we cannot make this statement. We have rewritten our conclusion to further reduce the potential for overinterpreting our findings: “Our findings relating Eng-implant with HIV susceptibility markers are subtle with unclear clinical impact, and consistent with the results ECHO trial findings.” (Page 17)

Reviewer #3: This article describes the results of a prospective, observational study of changes in HIV target cells and cytokine concentrations associated with use of the ENG contraceptive implant. Few data have been published on the association of ENG implant use and HIV risk to date. With implant use increasing rapidly in areas of high HIV prevalence (most notably in sub-Saharan Africa), the study addresses an important gap in the scientific literature on ENG implant use and HIV susceptibility. In their analysis, the authors evaluated changes in immunological markers in cervicovaginal fluid and blood PBMCs and identified associations between ENG implant use and increases in the proportion of CD4+ cells expressing CCR5 but decreases in the CD4:CD8 T cell ratio in the genital tract only. Results of analyses assessing markers of T cell activation were inconsistent in both tissue types and few changes in cytokine concentrations were identified after ENG implant initiation. This study’s strengths lie in its prospective design which accounts for potential variation in immunological markers at different stages of the menstrual cycle and adequate ascertainment and control of important covariates (STIs, BV). The study is limited by its small sample size as only 15 participants had data from all four pre/post implant study visits. Given this limitation and the fact that these data are drawn from an observational design, which could further be confounded by unmeasured factors, results should be interpreted carefully.

We agree with your last comment. To highlight this point we have specifically added the following sentence to our discussion. “Given our small sample size, these results should be interpreted cautiously” (Page 16, 6th paragraph)

Major comments

• Given the observational nature of this study, the authors should avoid causal language in the interpretation of their results, such as how implant use had an “effect on”, “results in” or “led to” changes in immunological markers. It’s highly recommended that this language be amended throughout the paper.

We have altered the language to reduce causal language

• Additionally, caution should be taken when interpreting changes in immunological markers as changes in HIV risk. Based on the final two sentences in the first paragraph of the discussion, it sounds as if the authors are suggesting that counseling around implant use should be altered based on the results of this study. This may be a step too far without evidence linking implant use with HIV acquisition. If anything, these results suggest that implant use may alter immunologic factors that have been shown to be associated with HIV risk, and highlight the need for further evidence. Please consider revising.

We have revised the text to reduce overinterpretation of the findings while highlighting the need for research in this area.

• The introduction could benefit from a clearer statement of the authors’ hypotheses and the biological or epidemiological evidence to support these hypotheses. Is there reason to believe from these studies that ENG exposure would increase HIV risk or alter immunity in the genital tract? Why is this research question worth pursuing aside from the current lack of evidence?

We recognize that our hypothesis was not clearly stated. We have added to our introduction “The Eng-Implant is a long-acting highly effective progestin-only contraceptive method containing a 3rd generation progestin. Although Eng-Implant has less glucocorticoid activity compared to medroxyprogesterone in DMPA and less endogenous estrogen inhibition, we hypothesize that given the sustained progestin exposure over time, there will still be some immunologic changes within the genital tract to suggest increased susceptibility with use.” (Page 4, 3rd paragraph)

• Clarification of the sampling is needed in the methods. Were all implant users from the parent study included? If not, how were ENG implant users sampled and is there reason to believe there is selection bias from the sampling?

All implant users from the parent study were included. Clarification to methods has been added. (Page 6)

• Please provide rationale for the decision to impute the lowest measured value for BLQ results rather than using an approach that is independent of measured values (e.g half of the LLOQ). Would this not bias your results?

We apologize for not including this detail. For samples below the level for quantification, we used half the lower limit of detection. (Page 10)

Minor comments

• In the introduction, it is worth noting that all of the studies that were conducted prior to ECHO were observational

This has been added (Page 3, 1st paragraph introduction)

• In the Results, consider adding the median/range of number of days from LMP that samples were collected for each pre-implant time point, and median/range number of days from implant insertion for the two post-implant time points. It is unclear whether you were able to collect the samples within the defined target luteal/follicular periods.

All pre-contraceptive samples were collected during the appropriate window described in the study methods, specifically 17 days to onset of next menses and less than 14 days after first day of last menstrual period for luteal and follicular phases, respectively. Visit 3 was conducted at a median of 84 days (Q1: 83. Q3: 86.5 days) post-contraceptive initiation and Visit 4 was conducted at a median of 105 days (Q1:98, Q3: 111) post-contraceptive initiation. This detail has been added to the text. (Page 11, 1st paragraph of results)

 What was the purpose of the models assessing the association between implant status with the CVL visit characteristics? Please clarify.

We compared characteristics of the visits before and after implant use to ensure there were no other factors we needed to consider in our analysis that may be confounding our study findings. This has been added to methods (Page 10, 8th paragraph)

• In the models assessing the distribution of lymphocyte memory cells, a linear model seems inappropriate with a categorical outcome. Can you clarify what your dependent variable was in these models or further explain your rationale for the linear model?

The outcomes within each memory cell subtype were proportion of cells with that characteristic profile. This proportion was continuous. (Page 11)

• Could you please elaborate on the types of STIs diagnosed in the first paragraph of the Results?

We have added detail to describe the specific STIs diagnosed. (Page 8)

• Did occurrence of detecting <100 CD3+ T cells vary significantly by the four visit types? If so, explain how this might influence your results.

There was no difference in the occurrence of low cell counts by visit type. This has been clarified in the results (Page 11, 1st paragraph)

• It is interesting that only a decrease in GCSF was noted in the genital tract after implant initiation. A similar reduction in GCSF was also observed in plasma (although not significant after adjusting for multiple comparisons). Consider discussing the role of this cytokine in the discussion and how it may relate to ENG exposure from a biological perspective.

We have added the following paragraph to the discussion:

“ A decrease in GCSF was noted in the genital tract after implant initiation. A similar reduction in GCSF was also observed in plasma (although not significant after adjusting for multiple comparisons). This finding of a small yet significant reduction in G-CSF may signify an alternative pathway associated with altered HIV susceptibility through damaged mucosa. Granulocyte colony-stimulating factor (G-CSF) may induce an inflammatory reaction enhancing neutrophil function. With receptors on granulosa cells, G-CSF has been implicated in ovulation and thus could be downregulated in the setting of ovulation inhibition associated with implant use. GCSF is also associated with wound healing and has been associated with faster healing from genital ulcerations, G-CSF stimulates the proliferation and differentiation of cells that participate in acute and chronic inflammation and immune responses including mature leukocytes, macrophages, and dendritic cells. This potential mechanism of altered immune response should be further explored to determine if clinically significant.” (Page 14)

• In the third paragraph of the discussion, the sentence beginning “Progestins can regulate the expression of numerous genes involved in multiple cellular functions…” should have a citation.

Citations have been added with sentence altered to reflect these relate to gonadal hormones.

• Author group for citation 4 (ECHO paper) needs to be corrected

Citation corrected

• It would be helpful to spell out the memory cell classifications in Figures 1 and 2 either in the legend or the description

 This has been added (Page 24 and 26)

6. PLOS authors have the option to publish the peer review history of their article (what does this mean?). If published, this will include your full peer review and any attached files.

Do you want your identity to be public for this peer review? For information about this choice, including consent withdrawal, please see ourPrivacy Policy.

Reviewer #1: No

Reviewer #2: No

Reviewer #3: No

While revising your submission, please upload your figure files to the Preflight Analysis and Conversion Engine (PACE) digital diagnostic tool,https://pacev2.apexcovantage.com/. PACE helps ensure that figures meet PLOS requirements. To use PACE, you must first register as a user. Registration is free. Then, login and navigate to the UPLOAD tab, where you will find detailed instructions on how to use the tool. If you encounter any issues or have any questions when using PACE, please email us at figures@plos.org. Please note that Supporting Information files do not need this step.

Lisa Haddad et al. studied the effects of Nexplanon contraceptive implant use on vaginal and systemic immune parameters associated with HIV acquisition. This report is timely due to the current controversy about whether depoprovera (DMPA) and other types of hormonal contraception affect susceptibility to HIV-1 infection. Like DMPA, the active component in Nexplanon, etonogestrel (Eng), is a synthetic progestin that targets the glucocorticoid receptor in addition to the progesterone receptor. This is the first report on the potential effects of Eng on biological factors underlying HIV risk. The investigators used a similar approach to earlier investigations done by others on the effects of DMPA on vaginal and systemic immunity. It should be noted that these studies to date have not reached a consensus on the effects of DMPA on vaginal immunity. [ Byrne et al (2016) detected increased numbers and frequency of CCR5+ CD4+ T cells in endocervical cytobrush samples from women on DMPA, but detected a similar increase during the luteal phase of the menstrual cycle in women that did not use DMPA. They also reported no significant effect on 14 vaginal cytokine concentrations. Smith-McCune et al (2017) detected no increase in the number or frequency of CCR5+ CD4+ T cells in endocervical cytobrush samples, but found increased concentrations of MCP1 and IFN alpha 2 in endocervical mucus. Mitchell et al (2014) found no effect of DMPA on numbers or frequency of CCR5+ CD4+ cells in ectocervical biopsies. Other large cohort studies have detected higher concentrations of RANTES and other chemokines in vaginal fluid from women on DMPA,]

This study reports a significant increase in the percent of T cells expressing CCR5 and the central memory phenotype (CCR7) in cells recovered from cervicovaginal lavage (CVL) fluid of women using Eng implants. 

Unlike previous studies, Haddad et al. used lymphocytes recovered from cervicovaginal lavage (CVLs) samples for phenotyping. This is a major limitation because lymphocytes in CVL samples are usually present in low in numbers and have poor viability. In this study, samples with fewer than 100 CD3+ T cells (20%) were excluded from analysis, but actual cell counts were not provided. Flow analysis is not accurate for samples with low cell counts (<20,000 cells). CVL mononuclear cell counts should be provided as the number of HIV target cells is a more meaningful endpoint than % HIV target cells. 

As noted above, we have added detail in our methods section to highlight why we selected CVL for our analyses. Further we have added in cell counts and cytokine concentrations in the tables. We believe however that the proportion expressing these markers are more appropriate method of evaluation and have maintained our analyses as such. In our discussion, we comment on this limitation. We do however believe that proportion of cells expressing different phenotypic markers highlight the characteristic quality of the immune response. (Pages 22, 23 and 29)

 The investigators reported collecting vaginal samples during the luteal and follicular phase of the menstrual cycle which is important because Byrne et al. reported a significant increase in CCR5+ CD4+ cells during the luteal phase. Did the investigators determine whether this was the case in their study before they combined these samples for the analysis? 

We did compare the Luteal and follicular findings and found no significant differences at these time points. Importantly our sample may not have been large enough to find such a difference, as possibly smaller than the effect of the implant. Notably, if there were a difference and we combined the groups, this would have biased our findings towards the null hypothesis. We have added this to our results and discussion sections (Page 11, 1dt paragraph, page 16, 5th paragraph)

Do women using Eng implants have cycling levels of endogenous progesterone and estrogen, and were these included as covariates in the analysis?

In this study we did not have endogenous hormonal concentrations available for comparison. We have added the following to the discussion: Although heterogeneity in the endogenous hormonal response to the contraceptive is possible and we did not measure and control for endogenous hormonal levels, we selected to include 2 time points post initiation to help control for some of that variability (Page 17, 6th paragraph)

Concentrations of cytokines in CVL fluid were also studied, and a modest but statistically significant increase in sCD40L, and decreases in Il-12 and G-CSF were noted. These data were presented as relative percentages (before and after ENG use). It would be helpful to also have the cytokine concentrations to provide a context for the potential physiological significance of these findings.

The cytokine values have been added as appendix tables. (See appendix Tables)

Since vaginal immune parameters are highly variable and there is no clear pattern of differences in vaginal parameters among women using progestin injections or implants for contraception, the differences in vaginal environment variables described in this report that are attributed to Eng use should be interpreted with caution. Conclusions such as: “Eng implant use led to a moderate increase in the availability of HIV target cells in the genital tract” and “ It is unclear if these implant induced changes would be any less safe than other contraceptives with regard to HIV risk” should be reworded. 

We have changed the wording throughout the manuscript to minimize the perception of any conclusion regarding increased risk.

---

## [Decision Letter · Decision Letter 1]

21 Feb 2020

PONE-D-19-23634R1

Impact of etonogestrel implant use on T-cell and cytokine profiles in the female genital tract and blood

PLOS ONE

Dear Dr Haddad,

Thank you for submitting your manuscript to PLOS ONE. After careful consideration, we feel that it has merit but does not fully meet PLOS ONE’s publication criteria as it currently stands. Therefore, we invite you to submit a revised version of the manuscript that addresses the points raised during the review process.

We would appreciate receiving your revised manuscript by Apr 06 2020 11:59PM. To enhance the reproducibility of your results, we recommend that if applicable you deposit your laboratory protocols in protocols.io, where a protocol can be assigned its own identifier (DOI) such that it can be cited independently in the future. For instructions see: http://journals.plos.org/plosone/s/submission-guidelines#loc-laboratory-protocols

We look forward to receiving your revised manuscript.

Kind regards,

Manish Sagar, MD

Academic Editor

PLOS ONE

Additional Editor Comments (if provided):

Editor’s comments

1) Please update this link: www.mtnstopshiv.org/node/773.

2) Please provide a citation for the following statement: “These CVL enriched lymphocytes are phenotypically and functionally shown to be comparable to cells resident at the underlying tissue.”

3) Please clarify are these only live cells, as assessed by Zombie staining: “CVLs collected from 53 (79% of all visits) visits contained greater than 100 CD3+ T-cells and were subsequently included in this analysis.”

4) In Table 3 please correct: % of CD4+ CCR5% T-cells expressing: to % of CD4+ CCR5+ T-cells expressing

5) Given the reviewer’s comments, please include the total number of viable CD4+ T cells at the different visits in Table 2.

Reviewers' comments:

Reviewer's Responses to Questions

**Comments to the Author**

1. If the authors have adequately addressed your comments raised in a previous round of review and you feel that this manuscript is now acceptable for publication, you may indicate that here to bypass the “Comments to the Author” section, enter your conflict of interest statement in the “Confidential to Editor” section, and submit your "Accept" recommendation.

Reviewer #1: (No Response)

Reviewer #2: (No Response)

2. Is the manuscript technically sound, and do the data support the conclusions?

Reviewer #1: (No Response)

Reviewer #2: Partly

3. Has the statistical analysis been performed appropriately and rigorously? 

Reviewer #1: (No Response)

Reviewer #2: Yes

4. Have the authors made all data underlying the findings in their manuscript fully available?

Reviewer #1: (No Response)

Reviewer #2: Yes

5. Is the manuscript presented in an intelligible fashion and written in standard English?

Reviewer #1: (No Response)

Reviewer #2: Yes

6. Review Comments to the Author

Reviewer #1: For the Figures 1 and 2, what’s the dependent variable in the linear mixed model? Please provide the model structure of the equation.

Reviewer #2: I appreciate that the authors, in response to reviewers’ comments, provided more details about the limitations of their study, provided more rigorous statistical analyses, and downplayed the potential implications of their results for HIV transmission.

However, I don’t think they adequately addressed the reviewers concerns about the use of CVL samples as a source of genital tract lymphocytes. The two references that were added to bolster the claim that CVLs provide adequate samples (Iyer SS 2017, Swaims-Kolmeier 2016), and others (McKinnon LR 2014, Archary D 2015) clearly indicate that the cell yield in CVLs is very low compared with other sampling techniques (cytobrush, biopsy). In McKinnon, the median CD4 cell count in CVLs was 89 vs. 1,170 for cytobrush samples. Furthermore, I did not find references for the claims (page 17) that cells in CVLs are phenotypically and functionally comparable to resident cells in underlying tissue; there are few CD4 T cells in the normal vaginal epithelium (Pudney J 2005), and cell viability is often an issue with CVL samples because the acidic vaginal pH can kill lymphocytes in minutes (Olmsted SS 2005). These issues remains a major weakness of this paper.

7. PLOS authors have the option to publish the peer review history of their article (what does this mean?). If published, this will include your full peer review and any attached files.

Reviewer #1: No

Reviewer #2: No

---

## [Author Response · Author response to Decision Letter 1]

27 Feb 2020

Editor’s comments

1) Please update this link: www.mtnstopshiv.org/node/773.

We have changed this link to reference a video of the CVL procedure used: https://vimeo.com/224957115/00cb72fed6

2) Please provide a citation for the following statement: “These CVL enriched lymphocytes are phenotypically and functionally shown to be comparable to cells resident at the underlying tissue.”

We have changed the sentence slightly, expanded the text to further clarify the role of luminal cells and added several citations to support these statements. “CVL or luminal cells are not imbedded within the tissue, persist within a harsh environment, and have a reduced cell yield compared with other sampling approaches, but CVL provides an accurate means of tissue resident phenotyping at the site of sexually transmitted infection exposure. In experiments where luminal T cells are analyzed separately from T cells embedded in the tissue, these two populations have been shown to be very similar phenotypically and functionally. 1-6 Microscopically, it has been shown that luminal T cells remain closely associated with the apical face of the epithelium. 7-11Several studies have shown these luminal T cells are viable, capable of recognizing and responding to antigen, and play a critical role in immunity at mucosal sites3,10,12-16. Luminal T cells are sufficient to provide significant protection even when T cells located in the underlying tissues are not present16, thus although you may not find a large number of T cells in CVL, these cells can be critical for barrier protection”.

3) Please clarify are these only live cells, as assessed by Zombie staining: “CVLs collected from 53 (79% of all visits) visits contained greater than 100 CD3+ T-cells and were subsequently included in this analysis.”

Yes, we performed and gated for cell viability following discrimination of single cell lymphocytes using the Biolegend Zombie Yellow™ Fixable Viability Kit as mentioned in the methods thus only considered viable leukocytes for the analyses. We adjusted the above to add that these contained greater than 100 viable CD3+ T-cells.

4) In Table 3 please correct: % of CD4+ CCR5% T-cells expressing: to % of CD4+ CCR5+ T-cells expressing

Thank you for noting this. Correction has been made in both Table 3 and Table 4

5) Given the reviewer’s comments, please include the total number of viable CD4+ T cells at the different visits in Table 2.

We have added this to table 2.

Reviewers' comments:

Reviewer's Responses to Questions

Comments to the Author

1. If the authors have adequately addressed your comments raised in a previous round of review and you feel that this manuscript is now acceptable for publication, you may indicate that here to bypass the “Comments to the Author” section, enter your conflict of interest statement in the “Confidential to Editor” section, and submit your "Accept" recommendation.

Reviewer #1: (No Response)

Reviewer #2: (No Response)

2. Is the manuscript technically sound, and do the data support the conclusions?

Reviewer #1: (No Response)

Reviewer #2: Partly

3. Has the statistical analysis been performed appropriately and rigorously? 

Reviewer #1: (No Response)

Reviewer #2: Yes

4. Have the authors made all data underlying the findings in their manuscript fully available?

Reviewer #1: (No Response)

Reviewer #2: Yes

5. Is the manuscript presented in an intelligible fashion and written in standard English?

Reviewer #1: (No Response)

Reviewer #2: Yes

6. Review Comments to the Author

Reviewer #1: For the Figures 1 and 2, what’s the dependent variable in the linear mixed model? Please provide the model structure of the equation.

The dependent variable is (continuous) percentage of cells in each mutually exclusive memory cell class. 

Reviewer #2: I appreciate that the authors, in response to reviewers’ comments, provided more details about the limitations of their study, provided more rigorous statistical analyses, and downplayed the potential implications of their results for HIV transmission.

However, I don’t think they adequately addressed the reviewers concerns about the use of CVL samples as a source of genital tract lymphocytes. The two references that were added to bolster the claim that CVLs provide adequate samples (Iyer SS 2017, Swaims-Kolmeier 2016), and others (McKinnon LR 2014, Archary D 2015) clearly indicate that the cell yield in CVLs is very low compared with other sampling techniques (cytobrush, biopsy). In McKinnon, the median CD4 cell count in CVLs was 89 vs. 1,170 for cytobrush samples. Furthermore, I did not find references for the claims (page 17) that cells in CVLs are phenotypically and functionally comparable to resident cells in underlying tissue; there are few CD4 T cells in the normal vaginal epithelium (Pudney J 2005), and cell viability is often an issue with CVL samples because the acidic vaginal pH can kill lymphocytes in minutes (Olmsted SS 2005). These issues remains a major weakness of this paper.

Thank you for this comment. We have expanded our response as noted above and hope this is now adequate. “CVL or luminal cells are not imbedded within the tissue, persist within a harsh environment, and have a reduced cell yield compared with other sampling approaches, but CVL provides an accurate means of tissue resident phenotyping at the site of sexually transmitted infection exposure. In experiments where luminal T cells are analyzed separately from T cells embedded in the tissue, these two populations have been shown to be very similar phenotypically and functionally. 1-6 Microscopically, it has been shown that luminal T cells remain closely associated with the apical face of the epithelium. 7-11Several studies have shown these luminal T cells are viable, capable of recognizing and responding to antigen, and play a critical role in immunity at mucosal sites3,10,12-16. Luminal T cells are sufficient to provide significant protection even when T cells located in the underlying tissues are not present16, thus although you may not find a large number of T cells in CVL, these cells can be critical for barrier protection”. 

7. PLOS authors have the option to publish the peer review history of their article (what does this mean?). If published, this will include your full peer review and any attached files.

Do you want your identity to be public for this peer review? For information about this choice, including consent withdrawal, please see our Privacy Policy.

Reviewer #1: No

Reviewer #2: No

While revising your submission, please upload your figure files to the Preflight Analysis and Conversion Engine (PACE) digital diagnostic tool, https://pacev2.apexcovantage.com/. PACE helps ensure that figures meet PLOS requirements. To use PACE, you must first register as a user. Registration is free. Then, login and navigate to the UPLOAD tab, where you will find detailed instructions on how to use the tool. If you encounter any issues or have any questions when using PACE, please email us atfigures@plos.org. Please note that Supporting Information files do not need this step.

1. Kohlmeier JE, Miller SC, Woodland DL. Cutting edge: Antigen is not required for the activation and maintenance of virus-specific memory CD8+ T cells in the lung airways. J Immunol 2007;178:4721-5.

2. Richter MV, Topham DJ. The alpha1beta1 integrin and TNF receptor II protect airway CD8+ effector T cells from apoptosis during influenza infection. J Immunol 2007;179:5054-63.

3. Bivas-Benita M, Gillard GO, Bar L, et al. Airway CD8(+) T cells induced by pulmonary DNA immunization mediate protective anti-viral immunity. Mucosal Immunol 2013;6:156-66.

4. Ye F, Turner J, Flano E. Contribution of pulmonary KLRG1(high) and KLRG1(low) CD8 T cells to effector and memory responses during influenza virus infection. J Immunol 2012;189:5206-11.

5. Macdonald DC, Singh H, Whelan MA, et al. Harnessing alveolar macrophages for sustained mucosal T-cell recall confers long-term protection to mice against lethal influenza challenge without clinical disease. Mucosal Immunol 2014;7:89-100.

6. Slutter B, Pewe LL, Kaech SM, Harty JT. Lung airway-surveilling CXCR3(hi) memory CD8(+) T cells are critical for protection against influenza A virus. Immunity 2013;39:939-48.

7. Wands JM, Roark CL, Aydintug MK, et al. Distribution and leukocyte contacts of gammadelta T cells in the lung. J Leukoc Biol 2005;78:1086-96.

8. Nakanishi Y, Lu B, Gerard C, Iwasaki A. CD8(+) T lymphocyte mobilization to virus-infected tissue requires CD4(+) T-cell help. Nature 2009;462:510-3.

9. Wu T, Hu Y, Lee YT, et al. Lung-resident memory CD8 T cells (TRM) are indispensable for optimal cross-protection against pulmonary virus infection. J Leukoc Biol 2014;95:215-24.

10. Laidlaw BJ, Zhang N, Marshall HD, et al. CD4+ T cell help guides formation of CD103+ lung-resident memory CD8+ T cells during influenza viral infection. Immunity 2014;41:633-45.

11. Hu Y, Lee YT, Kaech SM, Garvy B, Cauley LS. Smad4 promotes differentiation of effector and circulating memory CD8 T cells but is dispensable for tissue-resident memory CD8 T cells. J Immunol 2015;194:2407-14.

12. Hogan RJ, Zhong W, Usherwood EJ, Cookenham T, Roberts AD, Woodland DL. Protection from respiratory virus infections can be mediated by antigen-specific CD4(+) T cells that persist in the lungs. J Exp Med 2001;193:981-6.

13. Ely KH, Roberts AD, Woodland DL. Cutting edge: effector memory CD8+ T cells in the lung airways retain the potential to mediate recall responses. J Immunol 2003;171:3338-42.

14. Kohlmeier JE, Cookenham T, Roberts AD, Miller SC, Woodland DL. Type I interferons regulate cytolytic activity of memory CD8(+) T cells in the lung airways during respiratory virus challenge. Immunity 2010;33:96-105.

15. Horvath CN, Shaler CR, Jeyanathan M, Zganiacz A, Xing Z. Mechanisms of delayed anti-tuberculosis protection in the lung of parenteral BCG-vaccinated hosts: a critical role of airway luminal T cells. Mucosal Immunol 2012;5:420-31.

16. McMaster SR, Wilson JJ, Wang H, Kohlmeier JE. Airway-Resident Memory CD8 T Cells Provide Antigen-Specific Protection against Respiratory Virus Challenge through Rapid IFN-gamma Production. J Immunol 2015;195:203-9.

---

## [Editor Report · Decision Letter 2]

3 Mar 2020

Impact of etonogestrel implant use on T-cell and cytokine profiles in the female genital tract and blood

PONE-D-19-23634R2

Dear Dr. Haddad,

We are pleased to inform you that your manuscript has been judged scientifically suitable for publication and will be formally accepted for publication once it complies with all outstanding technical requirements.

With kind regards,

Manish Sagar, MD

Academic Editor

PLOS ONE
---

## [Editor Report · Acceptance letter]

12 Mar 2020

PONE-D-19-23634R2 

Impact of etonogestrel implant use on T-cell and cytokine profiles in the female genital tract and blood 

Dear Dr. Haddad:

I am pleased to inform you that your manuscript has been deemed suitable for publication in PLOS ONE. Congratulations! Your manuscript is now with our production department. 

With kind regards,

on behalf of

Dr. Manish Sagar 

Academic Editor

PLOS ONE